# Flexing with lines or pipes: Techno-economic comparison of renewable electricity import options for European research facilities

**Johannes Hampp** [ORCID]*

Center for International Development and Environmental Research, Justus Liebig University Giessen, Giessen, Hesse, Germany

* johannes.hampp@zeu.uni-giessen.de

## Abstract

Where local resources for renewable electricity are scarce or insufficient, long-distance electricity imports will be required in the future. Even across long distances, the variable availability of renewable energy sources needs to be managed for which dedicated storage options are usually considered. Other alternatives could be demand-side flexibility and concentrated solar power with integrated thermal energy storage. Here their influence on the cost of imported electricity is explored. Using a techno-economic linear capacity optimization, exports of renewable electricity from Morocco and Tunisia to CERN in Geneva, Switzerland in the context of large research facilities are modeled. Two different energy supply chains are considered, direct imports of electricity by HVDC transmission lines, and indirect imports using H2 pipelines subsequent electricity generation. The results show that direct electricity exports ranging from 58 EUR/MWh to 106 EUR/MWh are the more economical option compared to indirect H2-based exports ranging from 157 EUR/MWh to 201 EUR/MWh. Both demand-side flexibility and CSP with TES offer significant opportunities to reduce the costs of imports, with demand-side flexibility able to reduce costs for imported electricity by up to 45%. Research institutions in Central Europe could initiate and strengthen electricity export-import partnerships with North Africa to take on a leading role in Europe's energy transition and to secure for themselves a long-term, sustainable electricity supply at plannable costs.

## Introduction

Research facilities and communities are striving to become more sustainable, with a focus on reducing their carbon emission footprint. Among those working towards this goal are the High-Energy Physics (HEP) community and the research facilities at the European Organization for Nuclear Research (CERN) in Geneva, Switzerland [1]. A significant contribution to the carbon footprint of research institutions is made by electricity-related scope 2 emissions [2] from the use of fossil fuels in electricity generation. To some extent, these emissions can be lowered by increasing energy efficiency, but any remaining emissions have to be addressed by making the electricity supply carbon-neutral. This may be achieved by displacing fossil fuels in

**Data Availability Statement:** Results and model input data are available from Zenodo: https://doi.org/10.5281/zenodo.7623943 Model source code and workflow are available via the Software Heritage Archive: https://archive.softwareheritage.

org/browse/directory/eb710eb81a71e422
c072330b77c55668273ea6fa/?origin_url=https://
github.com/euronion/trace& revision=
95849ab90ae969b8c0b2a238b8a8c2
37ec56d230&snapshot=7c3aa9b2ed601c
'094344ae4a2630e7f9160202d3.

**Funding:** The author(s) received no specific
funding for this work.

**Competing interests:** The authors have declared
that no competing interests exist.

**Abbreviations: CCGT**, Closed-Cycle Gas Turbine;
**CERN**, European Organization for Nuclear
Research; **CSP**, Concentrated Solar Power; **EAC**,
Equivalent Annual Cost; **ESC**, Energy Supply Chain;
**FOM**, Fixed Operation & Maintenance; **GEGIS**,
GlobalEnergyGIS; **H$_2$**, Hydrogen; **HEP**, High-Energy
Physics; **HVAC**, High-Voltage Alternating Current;
**HVDC**, High-Voltage Direct Current; **LCoE**,
Levelised Cost of Electricity; **LHC**, Large Hadron
Collider; **MA**, Morocco; **PV**, Photovoltaic; **PyPSA**,
Python for Power System Analysis; **RES**,
Renewable Energy Source; **TES**, Thermal Energy
Storage; **TN**, Tunisia; **UK**, United Kingdom; **WACC**,
Weighted Average Cost of Capital.

electricity generation with low-emission alternatives like Renewable Energy Sources (RES)
from wind and solar technologies, primarily Photovoltaic (PV). However, expanding RES
capacities becomes increasingly difficult with increasing shares of RES in the electricity mix
due to their land requirements and weather-dependent availability needing to be managed. In
Central Europe, both pose major challenges because of high population density, low land avail-
ability [3], and strong seasonality of RES.

One strategy to address the challenges is to deploy RES in areas with lower population den-
sity and more favorable climatic conditions, for example, solar-dependent RES closer to the
equator, where diurnal and seasonal variations are reduced [4]. In several cases, Northern Afri-
can countries including Morocco (MA) and Tunisia (TN) have been considered in the past as
locations to produce electricity using RES and export it to European countries, see for example
[5–8]. One initiative which became widely known in the 2000s was DESERTEC with plans to
import electricity from Northern Africa to Europe [9]. A more recent example is the company
Xlinks, which announced in 2021 its plans for the "Morocco-UK power project", a project to
export electricity from MA to United Kingdom (UK) [10]. The project aims for building RES
capacities in MA, a mix of wind and solar PV of a total of 10.5GW capacity, combined with
20GWh battery storage for managing their weather-dependent availability. The electricity is
intended to be exported by subsea High-Voltage Direct Current (HVDC) transmission line
with a capacity of 3.6GW along the Western European continental coast to Devon, UK. Other
underwater, long-distance HVDC transmission line projects are under construction or
planned in Europe with its neighboring continents [5], like the EuroAsia interconnector [11],
showcasing the technical and economic feasibility of such projects.

In the realization of large RES projects, reliable partnerships and off-take guarantees for the
imported electricity are essential, as these reduce a project's risks and improve financing condi-
tions through reduced cost of capital [12]. Through favorable financing conditions, RES-based
projects can become economically feasible and attractive compared with fossil-based energy
projects, as they are typically characterized by high upfront investment and low operational
costs as opposed to fossil-based projects with low investment and high operational costs [13].

Herein lies an opportunity for synergies between research institutions and potential energy
import projects to be explored. CERN has a high electricity demand, namely around
1250GWh in 2018 during "Run 2" of the Large Hadron Collider (LHC) [14]. Like other large
consumers of electricity, low and stable electricity prices are important for CERN and its cur-
rent as well as future operations. At the moment CERN is supplied through the French elec-
tricity grid [14], which had a low carbon intensity 67 $67g_{CO_2}$/kWh in 2018 compared to other
European countries, due to France's high share of nuclear power [15] in its electricity supply.
More recently the French nuclear power plant fleet has been experiencing a rising number of
problems. And with the fleet's future unclear, maintaining current levels of carbon emissivity
and electricity prices in the French grid will become impossible without the addition of signifi-
cant RES capacities [16]. CERN, as an international research institution, could use the oppor-
tunity to pursue an import project for RES-based electricity from a neighbor country outside
Europe, securing for itself and other interested research institutions a supply of electricity at
long-term stable and plannable costs, thereby reducing the project risks and increasing the
chances of the project being realized and adding to the French and European energy
transition.

In assessing the viability of such a potential project, techno-economic modeling and analy-
sis play a crucial role to understand associated limitations as well as technical and economic
details. There exists a large body of literature on techno-economic analysis for projects export-
ing energy from Northern Africa to Europe with different foci. Electricity has been

investigated with great interest in the past, e.g. [17–20] and recently Hydrogen ($H_2$) has become increasingly discussed for the same purpose, e.g. [21–23]. A very limited number of studies include both electricity and hydrogen exports in their analysis, like van der Zwaan et al. (2021) [8], and thereby allowing for comparative insight into these two alternatives.

While it is recognized that techno-economic modeling which includes RES needs to account for the variable availability of RES, e.g. through a sufficiently high temporal resolution [24], fewer analyses explore how this variability is managed. One common approach is the use of one or a combination of storage technologies [25]. The other is demand-side flexibility, which despite being a promising alternative [26], is often neglected [27]. With demand-side flexibility, the electricity demand is shifted in time based on the availability of RES-based electricity supply.

This study addresses these two common shortcomings—a comparison between hydrogen and electricity and demand-side flexibility as an alternative to storage—and compares the costs of imported electricity, RES shares, and storage requirements for large-scale electricity exports by HVDC and $H_2$ pipeline from MA and TN to Central Europe.

## Method

### General

The methodology for this paper is based on Hampp et al. (2023) [28] with some modifications. The investigated import projects, from now on referred to as Energy Supply Chains (ESCs), are modeled based on their techno-economic properties. For each ESC and exporter, an independent green-field capacity expansion model is built, where the model determines technical capacities to meet the requested electricity demand under optimal annualized system costs and perfect foresight in a linear program. Annualised costs $c_i$ for each component $i$ are calcualted using the Equivalent Annual Cost (EAC) method:

$$c_i = C_i \cdot (A_i + \text{FOM}_i) \tag{1}$$

where $C_i$ represent the components investment and $\text{FOM}_i$ the Fixed Operation & Maintenance costs. $A_i$ is the annuity factor for the component with lifetime $t_i$ and cost of capital $r$:

$$A_i = \frac{(1+r)_i^t \cdot r}{(1+r)_i^t - 1} \tag{2}$$

To properly represent the variability of RES availability, a full year is modeled with an hourly time resolution. Python for Power System Analysis (PyPSA) [29] is used as an energy system modeling framework to implement the capacity expansion models. A major change to the original methodology from [28] is the substitution of the software package GlobalEnergy-GIS (GEGIS) [30] with *atlite* [31]. While both packages can be used for modeling availability time series for RES from weather data and to determine land-availability potentials, *atlite* has a more modular and extensible approach which is utilized in this study.

As import destination Geneva is selected as a proxy for Central Europe. In Geneva and the surrounding regions, CERN and other large European research institutions are situated. The location also features a well-connected grid infrastructure through which potential synergies the European electricity grid could be created.

### Renewable potentials and availability

As exporting countries MA and TN are chosen. Both countries are outside of but neighboring Europe in the Mediterranean and are thus in proximity to Geneva. As both countries are closer

to the equator, they experience lower seasonality and a longer time of diurnal sunshine duration for higher availability of solar-based RES, PV, and Concentrated Solar Power (CSP). Due to lower population densities in MA and TN compared to Central Europe, their land availability for RES is also more favorable. For both countries, their southernmost administrative regions are excluded to reduce the spatial spread of potential RES sites. Eligible areas for each RES technology are identified using GEBCO [32] for slope and sea depth, WDPA [33] for protected areas, Corine Land Cover [34] for land cover types and proximity restrictions to populated areas and IMF data [35] on shipping sea routes. The resulting eligible areas are included in S4 Appendix. Using the eligible areas, RES potentials are determined by assuming technical potentials of 3 MW/km$^2$ for offshore as well as onshore wind, and 17 MW/km$^2$ for solar PV and CSP, based on [36]. Weather data for modeling RES hourly availability using *atlite* [31] is based on reanalysis data from ERA5 [37] for 2013. For the conversion from weather data to RES availability different models are implemented in *atlite* and used here, namely for onshore a *Vestas V112 3MW* and for offshore wind a *NREL offshore 5-MW baseline wind turbine* are used. Solar PV is modeled using the crystalline silicon model by Huld et al. (2010) [38]. For CSP the field of solar tower power station is represented with a solar-position dependent field and tower irradiation-to-heat efficiency which emulates detailed system simulations using *NREL SAM* [39] in combination with a fixed-efficiency power block. All RES locations are classified based on their annual capacity factor into 50 quality classes for each technology to construct country-specific electricity supply curves. For each technology and quality class, an hourly availability time series is generated which is made available to the capacity expansion model. Local electricity demand in the exporting regions MA and TN are considered by removing the estimated amount of annual electricity demand from the lowest-cost resources in the respective annual supply curves, see [28] for details. The annual electricity demands for MA (122TWh) and TN (64TWh) are synthetic demand estimates for 2030, generated using *GlobalEnergyGIS* [30] with 2013 as a reference weather year and GDP and population growth based on the SSP2 [40] scenario.

## Technologies

As electricity generation technologies, (1) solar PV, (2) onshore wind and (3) offshore wind are included in all scenarios and ESCs. Furthermore in additional scenarios the influence of having (4) CSP with integrated Thermal Energy Storage (TES), as a special RES with an integrated storage option, as a fourth energy source available is analyzed.

Two types of ESCs are considered which both deliver electricity to the importer site in Geneva:

1. "HVDC" ESCs: Electricity is directly transported from the exporter to the importer by HVDC transmission line

2. "H$_2$ pipeline" ESCs: Electricity is first used to produce H$_2$ on the exporter side, and the H$_2$ is then transported by H$_2$ pipeline to the importer side where it is used for electricity generation using a Closed-Cycle Gas Turbine (CCGT).

The CCGT turbine for H$_2$-to-electricity conversion is chosen for its lower investment costs compared to large-capacity fuel cells. The ESCs and technologies are visualized in Fig 1. Technology costs and efficiencies are assumed for 2030 and listed in Table 1. Costs are assumed for large-scale facilities and scale linearly with capacity, no additional scaling exponent is applied. Efficiencies are assumed constant and load-independent. HVDC transmission lines and pipelines are transport modeled also with fixed, distance-dependent transport efficiencies. No standing losses for battery, hydrogen, and thermal energy storage are assumed. Technology

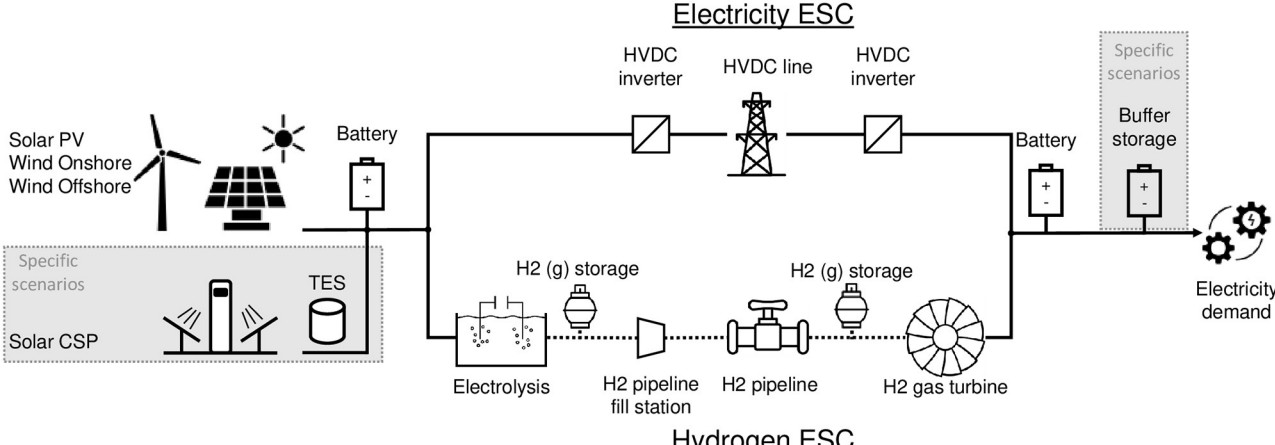

**Fig 1. Technologies used for modeling the electricity imports ESCs and during specific scenarios.** (Top) ESC shows the technologies involved with direct electricity imports by HVDC, (Bottom) shows technologies for indirect imports by H2 pipeline. Specific to certain scenarios is the inclusion of CSP with TES and the capacity of the electricity buffer storage for demand-side flexibility modeling. Icons are CC-BY-3.0 [41–47] and own creation.

costs and other currency values are reported in $EUR_{2015}$. Within the model technology costs are annualized using the equivalent annual cost method, considering technology investment and Fixed Operation & Maintenance (FOM) costs with 10% Weighted Average Cost of Capital (WACC). The method for calculating annualized component costs and a discussion of the choice of WACC are discussed in greater detail in [28].

Based on the assumptions made for conversion between electricity and $H_2$, a maximum round-trip efficiency for the $H_2$ ESCs of 39.44% can be established. When considering transport efficiencies and losses, a lower efficiency for the full $H_2$ ESCs can be expected.

**Table 1. Technology assumptions for the model for 2030.** Sources and a detailed description of the selected technologies are listed in S2 Appendix.

| Technology | Investment [$EUR_{2015}$] | | FOM [%/year] | Lifetime [year] | Efficiency [%] |
|---|---|---|---|---|---|
| Wind onshore | 1035.56 | $EUR/kW_e$ | 1.22 | 30 | - |
| Wind offshore | 1523.55 | $EUR/kW_e$ | 2.32 | 30 | - |
| Solar PV | 347.58 | $EUR/kW_e$ | 2.48 | 40 | - |
| CSP field & receiver tower | 98.15 | $EUR/kW_{th}$ | 1.1 | 30 | - |
| CSP TES | 13.15 | $EUR/kWh_{th}$ | 1.1 | 30 | - |
| CSP power block | 687.5 | $EUR/kW_e$ | 1.1 | 30 | 41.2 |
| Battery storage | 142 | $EUR/kWh_e$ | 0 | 25 | - |
| Battery inverter | 160 | $EUR/kW_e$ | 0.34 | 10 | 98 |
| Electrolysis (Alkaline) | 450 | $EUR/kW_e$ | 2 | 30 | 68 |
| Hydrogen storage tank | 12.23 | $EUR/kWh_{H2}$ | 2 | 20 | - |
| HVDC inverter pair | 162.36 | $EUR/kW_e$ | 2 | 40 | 98 |
| HVDC line overhead on-land | 432.97 | EUR/(MW km) | 2 | 40 | 97.7/1000km |
| HVDC line underwater | 471 | EUR/(MW km) | 0.35 | 40 | 97.7/1000km |
| $H_2$ (g) pipeline fill station | 4478 | $EUR/MW_{H2}$ | 1.7 | 20 | 97.9 |
| $H_2$ (g) pipeline on-land | 226.47 | EUR/(MW km) | 3.17 | 50 | 97.9/1000km |
| $H_2$ (g) pipeline underwater | 329.37 | EUR/(MW km) | 3 | 30 | 97.9/1000km |
| $H_2$ gas turbine (CCGT) | 830 | $EUR/kW_e$ | 3.35 | 25 | 58 |
| Demand-side flexibility (Buffer storage) | 0 | $EUR/MWh_e$ | - | - | - |

## Import routes

The same routes are used for HVDC transmission lines and $H_2$ pipelines from both of the exporting countries. With the applied methodology the exact starting locations of the export routes do not impact the results, here Fez, MA, and Kairouan, TN, are chosen due to their central locations and access to pre-existing infrastructure in their respective country. From the starting points, the export routes head towards the coast and cross the Mediterranean to follow along the coastline of either Spain or Italy. From MA the path follows along the Spanish coast mostly underwater to Montpellier in France and then via Lyon to Geneva. For TN the path connects to Sicily by underwater line and from there follows the Italian coastline on-land to northern Italy and then Geneva. These routes are chosen to follow the shallow (less than 400 m) coastline of Spain with undersea HVDC lines to reduce public resistance and complicated permitting, the same reason as the Xlinks project chooses an underwater connection instead of a shorter on-land connection. The comparable underwater route for Italy along the coastline would for most parts be below a sea depth of 800 m. For such depths few projects exist, therefore the route is avoided as it would pose additional technological challenges. Instead, the route for Italy is chosen to follow existing High-Voltage Alternating Current (HVAC) lines from southern to northern Italy. The resulting routes as shown are comparable to those presented by Hess (2018) [48] and should not be considered definite. To account for potential deviations from these routes, the route lengths are scaled by a detour factor of 1.2 for HVDC [49] and 1.4 for pipeline [21]. This factor also helps to account for collection infrastructure on the exporting sides of MA and TN, which is not considered separately. The routes for MA and TN both have a total length of 1900km. They are shown in detail in Fig 2 and the individual distances are given in Table 2.

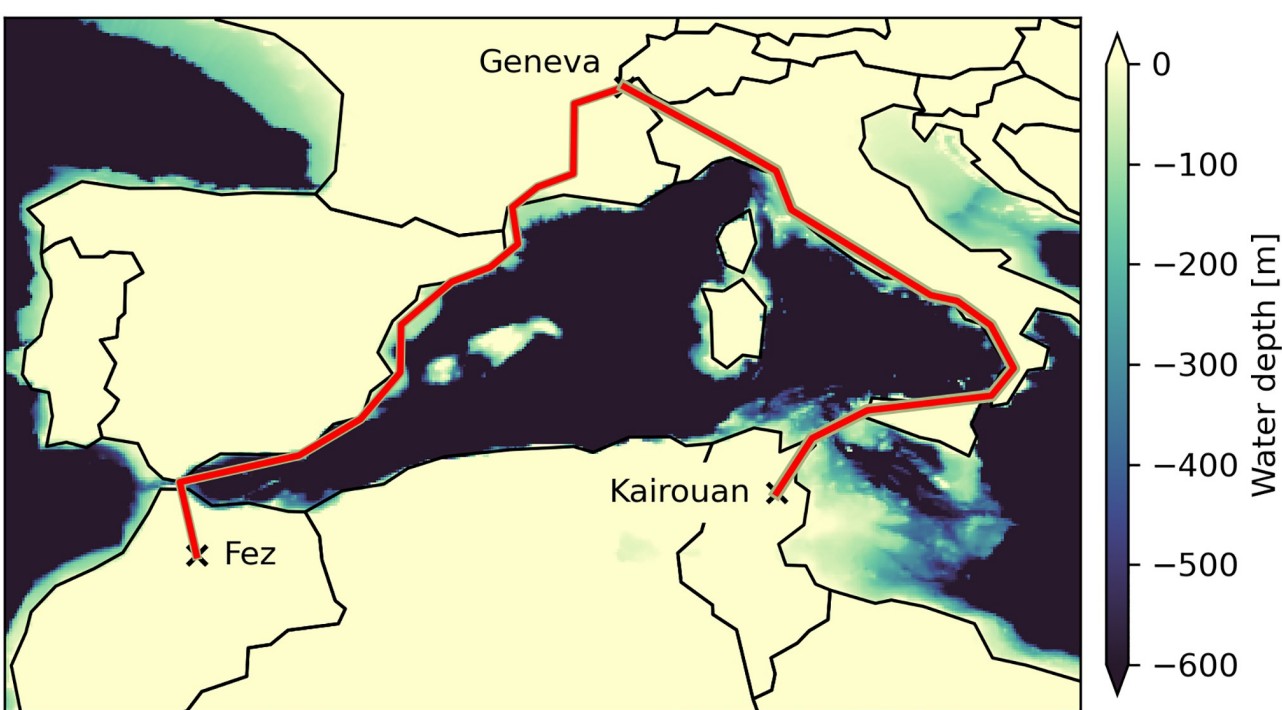

**Fig 2. Import routes from MA and TN to Geneva, Central Europe, which are selected for this study.** The route from Fez, Morocco, follows the Spanish coastline underwater. The route from Kairouan, Tunisia, follows the Italian coastline on-land as the sea depth would make underwater development challenging. Country shapes made with Natural Earth [50] data, water depth based on [32].

**Table 2. The distances considered for the different ESCs and exporters.** Total distances for both exporters are the same with different shares of distance for the on-land and underwater paths.

| Distances [km] | Total (underwater) | HVDC[a] | H₂ pipeline[b] |
|---|---|---|---|
| Morocco (MA) | 1900 (1380) | 2280 (1656) | 2660 (1932) |
| Tunisia (TN) | 1900 (150) | 2280 (180) | 2660 (210) |

[a] Distances scaled by a detour factor of 1.2 based on [49].

[b] Distances scaled by a detour factor of 1.4 based on [21].

### Export, technology and demand-sid flexibility scenarios

Imports from MA and TN to Geneva are compared between different scenarios:

- The two different ESCs, import by HVDC transmission line and H₂ pipeline

- With and without CSP & TES available as generation and storage technologies

- With different demand flexibilities for shifting the electricity demand pattern

The electricity demand in all scenarios is set to a constant baseload of 3.6GW to keep comparability with the Xlinks project.

Demand-side flexibility in the different scenarios refers to the ability to consume the requested amount of electricity at an earlier or later time. Increasing demand-side flexibility allows the model to meet electricity demand at an earlier or later time. This opens an additional option for the model to address the variable availability from the RES other than through storage capacities. The scenarios for demand-side flexibility range from "Baseload", i.e. the case where the electricity demand of 3.6 GW has to be continuously supplied, to "Annual" matching where there are no restrictions on when the electricity is supplied, only the total amount of 3.6 GW · 8760 h = 31 536 GWh needs to be supplied by the modeled ESCs. Scenarios with flexibility in shifting their demand, i.e. scenarios between "Baseload" and "Annual", allow the model to shift an electricity demand equivalent to the scenario name to an earlier or later time. The demand flexibility is implemented using an electricity buffer storage before the electricity consumer, which the model can use at no costs and with a maximum electricity storage capacity equivalent to the scenario name, e.g. for the "Daily" scenario with a capacity of 86.4 GWh = 3.6 GW · 24 h. Similarly, the remaining flexibility scenarios allow for demand shifting within "Weekly" for 168 h, "Biweekly" for 336 h, "Monthly" for 744 h and "Quarterly" for 2208 h. Technically the demand-side flexibility is implemented by extending the *PyPSA* models from [28] with a for-free *store*-component before the final electricity demand with its' maximum capacity limited to the amount of demand-side flexibility.

## Results

### Costs for importing electricity

The costs for importing electricity through the HVDC and H₂ pipeline ESCs are presented in Fig 3 for all scenarios modeled for this study, with both MA and TN as exporters and different levels of demand-side flexibilities. The figure shows the Levelised Cost of Electricity (LCoE), which is calculated as the total system costs divided by the annual total of electricity delivered. Tabular results are also included in S1 Appendix.

For both MA and TN exporters, the costs of importing electricity are similar. The costs of importing electricity directly through the HVDC ESC range from 58 EUR/MWh to 106 EUR/ MWh. For indirect imports through the H₂ pipeline ESC, the costs are more than twice as

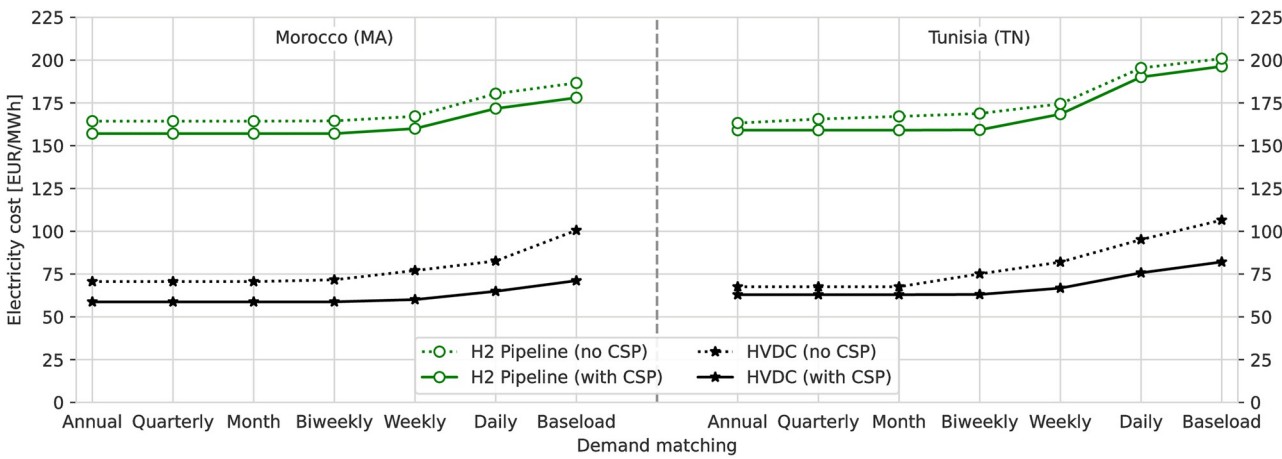

**Fig 3. Cost of direct and indirect imports electricity from MA (left) and TN (right) in all scenarios.** The results are qualitatively and quantitatively similar for MA and TN. Direct imports by HVDC are 2 to 3 times cheaper than indirect imports and power generation from H₂. Electricity costs are highest when baseload electricity is required. Using a combination of CSP with TES as additional RES can offer additional cost benefits.

high, starting from 157 EUR/MWh to 201 EUR/MWh. Scenarios with both CSP and TES for electricity generation and energy storage (referred to as "with CSP") have lower costs compared to their scenario counterparts where both technologies are not available (referred to as "no CSP"). In the case of HVDC ESCs, the cost benefits of having CSP are higher, reaching up to a reduction of 25 EUR/MWh or 25% in the "Baseload" scenario. On the other hand, the cost benefits of having CSP in H₂ ESCs are relatively low at the highest 10 EUR/MWh or 6%. More systematic cost benefits for scenarios of both ESCs can be found with increasing demand-side flexibility. Here the costs of all imports can be reduced by 20 EUR/MWh to 40 EUR/MWh or 20% when relaxing the requirement for "Baseload" to "Annual"-matched electricity supply. Since the costs and other results are very similar for MA and TN, for the remainder of this study the presentation will focus on the results for MA with and without CSP. Additionally, the focus will be on comparing "Baseload" and "Annual" scenarios, as they show the biggest differences and the results of the remaining flexibility scenarios are found to be these two scenarios. Figures showing the results for TN and all flexibility scenarios are included in S3 Appendix.

## Electricity generation mix

Electricity for direct export and H₂ production is procured from a mix of wind and solar RES. A major difference between the scenarios is the availability of CSP and TES for electricity generation. Figs 4 and 5 show the total electricity generation by technology for the HVDC and H₂ ESCs, comparing scenarios with and without CSP. RES capacities available for electricity generation but unutilized are shown as curtailed electricity. While the total amount of produced electricity is similar for all scenarios of the same ESC, the H₂ ESC requires more than double the amount of electricity compared to the HVDC ESC due to the round-trip conversion losses between electricity and H₂. Without transport energy demand and losses, the conversion of electricity to H₂ and back has a round-trip efficiency of 39.44%. The electricity is provided from a mix of solar PV (or CSP if available in the scenario), combined with small amounts of wind. Solar-based RES are at an advantage in Northern Africa due to the low seasonality and the diurnal availability pattern. This availability pattern allows PV to be combined with battery storage and CSP with TES to provide baseload electricity relatively easily.

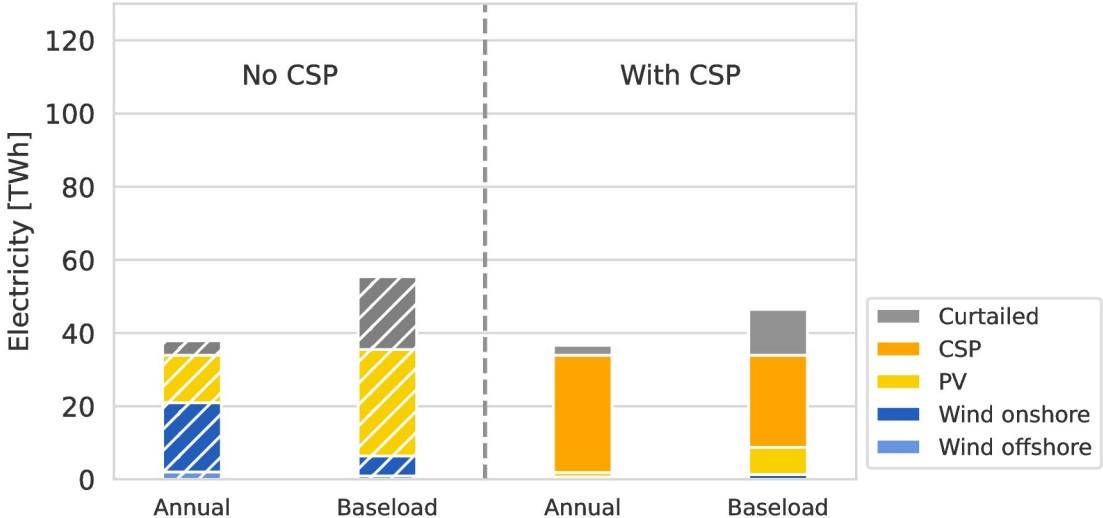

**Fig 4. Electricity generation mix for direct imports by HVDC from MA.** Scenarios without CSP and TES (left) compared to scenarios with both technologies available (right) for the two most contrasting scenarios for demand-side flexibility, "Baseload" and "Annual" matching, are compared. Without CSP and TES, electricity is generated from a mix of PV and onshore wind, where PV dominates in the "Baseload" scenario. In scenarios with CSP and TES, their combination leads CSP to be the dominant source of electricity, and the low-cost TES leads to lower levels of curtailed electricity.

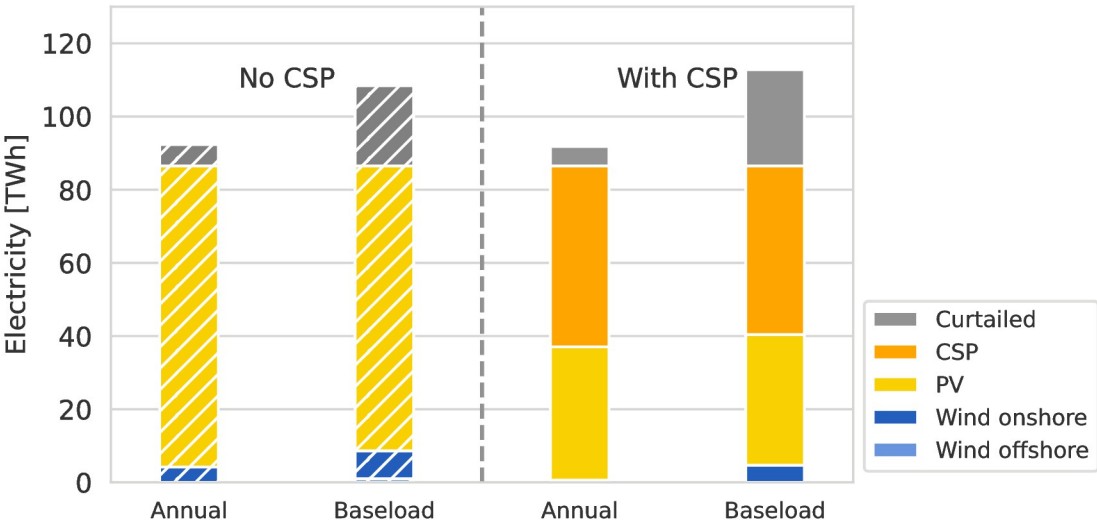

**Fig 5. Electricity generation mix for indirect imports by H$_2$ from MA.** Scenarios without CSP and TES (left) compared to scenarios with both technologies available (right) are compared. In contrast to Fig 4, PV generally provides a larger share of electricity in all scenarios and the total electricity generation is more than two times higher compared due to the round-trip-efficiency of converting between electricity and hydrogen.

## Storage and transport capacities

The capacities for storage, transport, and H$_2$ conversion infrastructure from MA for scenarios including CSP are shown in figure Fig 6. The capacities are reported in electricity equivalents (GWh$_e$, GW$_e$) for better comparison. This means that the H$_2$ storage and pipeline capacities are multiplied by the CCGT efficiency and TES capacities are multiplied by the CSP power block efficiency (cf. Table 1).

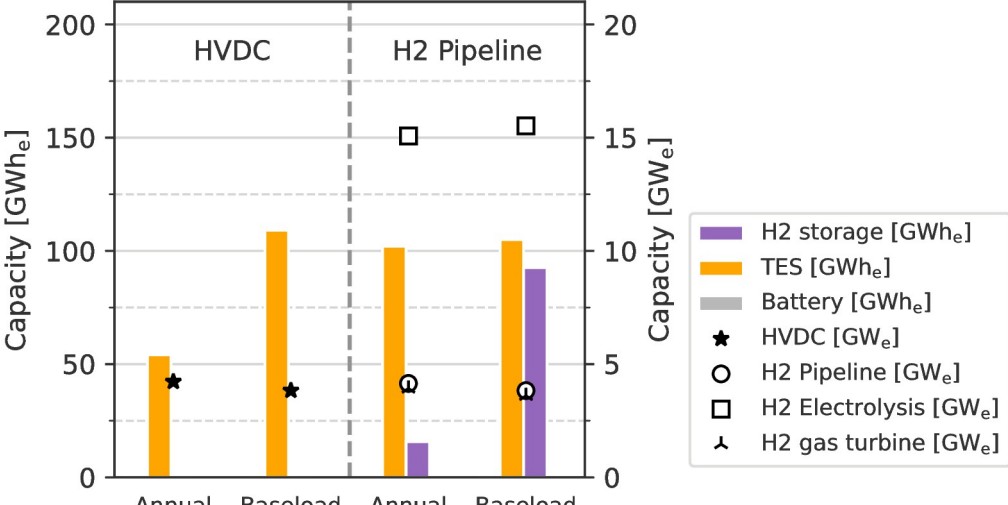

**Fig 6. Capacities for storage, transport, and H$_2$ technologies for (left) direct and (right) indirect imports from MA in scenarios with CSP and TES.** Storage capacities are in GWh$_e$ (bars, left y-axis), transport and H$_2$ technologies are in GW$_e$ (markers, right y-axis). All capacities are reported in electricity equivalents, see text for details. As demand becomes less flexible, storage capacities increase and are the highest in the scenarios for "Baseload" electricity supply.

The transport capacities for both HVDC and H$_2$ ESCs are similar across all scenarios, which indicates the limited surplus capacity and a high utilization rate for this infrastructure. The model has in principle the option to build storage capacities also after the transport infrastructure, that is downstream after the HVDC lines and H$_2$ pipeline on the exporting side. However, most storage is built on the exporting side in MA to maintain the high utilization rate of the transport infrastructure and to buffer the variable availability of RES there, rather than on the importing side.

The storage capacities of the H$_2$ ESC scenarios are about twice as high as those of the corresponding HVDC ESC scenarios. Between "Annual" and "Baseload" scenarios, the total storage capacities increase as the supply needs to meet demand on a stricter schedule. The overall storage capacities built range from 54 GWh$_e$ to 110 GWh$_e$ for direct imports via HVDC and 118 GWh$_e$ to 198 GWh$_e$ for indirect imports via H$_2$ pipeline. The storage capacities between the ESCs can also be compared based on how long they can provide the fixed demand of 3.6 GW. In the case of HVDC-based, the capacities last between 15 h to 31 h. For H$_2$ pipeline-based imports, the storage capacities last for a duration of 16 h to 37 h, the round-trip conversion to and from H$_2$ is accounted for the TES capacity, as the energy stored in the TES is exposed to conversion losses when being moved downstream along the H$_2$ ESC, cf. Fig 1.

As for the deployed technologies, battery storage is deployed in only some scenarios of the HVDC ESC and only when TES is not available, cf. S3 Appendix. In the scenarios as shown in Fig 6, where TES is available it fully replaces battery storage due to its lower costs. In the case of the H$_2$ ESC, the increasing storage capacity requirements are met with the help of H$_2$ storage, due to its even better economics and integration of bulk H$_2$ storage compared to the alternatives TES and battery storage. At the same time TES plays a smaller role in the overall storage capacity while playing an important role in improving the economics of H$_2$ electrolysis by increasing the electrolysis' utilization factor. The necessary electrolysis capacities for the H$_2$ ESC with more than 15GW at first glance seem high in comparison to the transport infrastructure capacity of below 4GW$_e$. As for the TES, the round-trip efficiency of 39.44% for H$_2$ conversion has to additionally be taken into account for the electrolysis capacities, leading to a

minimum requirement of $^{3.6GW}/_{39.44\%} = 9.13\text{GW}_e$. This gives a utilization rate of around 60% for the electrolysis in the shown scenarios. This utilization rate is well-above the possible utilization rate from a pure solar-based RES supply mix (cf. Fig 5) which is exposed to the day-night cycle. This points to the role of TES in the $H_2$ ESCs, as being to increase the utilization rate of electrolysis, rather than providing bulk energy storage capacities. This role is reserved for the $H_2$ storage.

### Import cost compositions

The import costs can be broken down into individual cost components, which is shown in Fig 7. Approximately half of the total costs are for electricity generation from RES. Between "Annual" and "Baseload" scenarios, there are two components responsible for driving costs up. The first is the increasing RES costs linked to increasing RES capacities (cf. Figs 4 and 5), and the second the increasing storage costs linked to their increasing capacities, required to provide the baseload electricity supply.

Between the HVDC and $H_2$ ESCs, the cost difference is related to $H_2$ production and electricity generation from $H_2$. RES costs are nearly double for the $H_2$ pipeline compared to the HVDC-based imports, as conversion losses along the ESC accumulate and need to be compensated by higher electricity generation. The $H_2$ electrolysis and electricity generation from $H_2$ by gas turbine add another 25% to the total costs and are additional cost components that are not required by the HVDC ESC. The comparison also shows that the transport-related costs, i.e. for HVDC transmission lines and $H_2$ pipeline including auxiliary equipment, are similar across all ESC and none of the alternatives has a unique advantage over the others. $H_2$ imports do not profit from lower transport costs.

### Discussion

The LCOE for importing electricity determined in this study are within the range of other studies. Compared to an earlier study by the author [28] where direct and indirect imports of

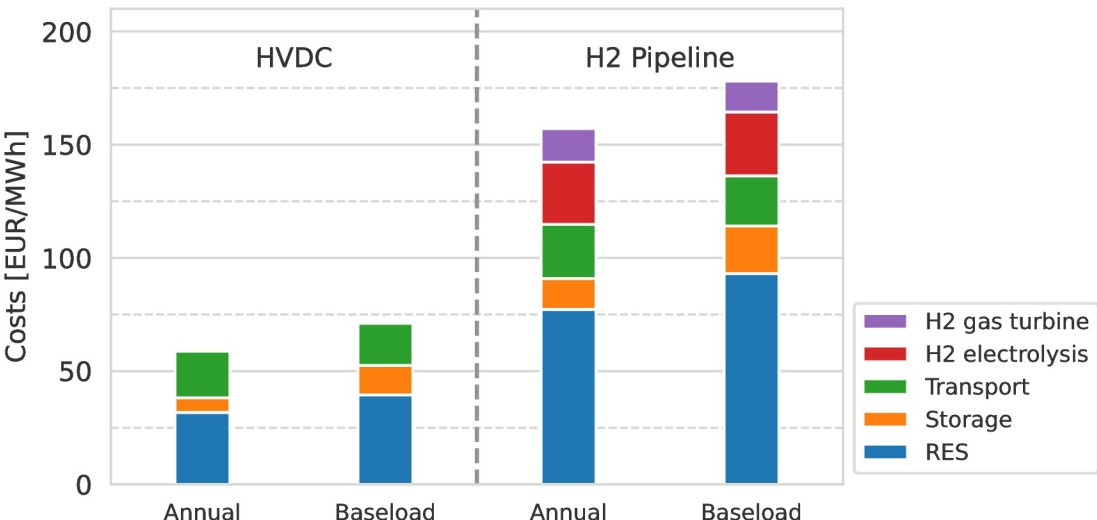

**Fig 7. Composition of import costs for (left) direct and (right) indirect imports from MA in scenarios with CSP and TES.** The costs for transport are similar across scenarios and both ESC. The H2 conversion steps add additional costs and electricity demand, leading to higher RES capacities required and RES-related costs. Between "Annual" and "Baseload" demand matching differences in costs are due to additional storage capacities and RES overcapacities, required for balancing weather-depended RES availability.

$H_2$ by pipeline and HVDC were compared, in this study the advantages of directly importing electricity as the demanded energy carrier are again underlined. Hess (2013) [51] estimated around 120 EUR/MWh for HVDC-based electricity imports without demand-side flexibility. For imports of $H_2$ via blending in natural gas pipelines Timmerberg et al. (2019) [21] estimated in low and high cost scenarios to be between 59 EUR/MWh$_{H2}$ to 123 EUR/MWh$_{H2}$, or 102 EUR/MWh to 212 EUR/MWh after taking into account the CCGT efficiency relevant in this study. The range found by Timmerberg et al. for low and high-cost scenarios is quantitatively similar to the range found here due owing to demand-side (in-)flexibility, indicating the importance of considering demand-side flexibility in such studies. The benefits of demand-side flexibility found here are also comparable to those found by Walter et al. (2023) [26] who determined possible reductions to costs for hydrogen of 30 EUR/MWh from full temporal demand-side flexibility.

Results based on the methodology and models used here are sensitive to made technology assumptions (cf. Table 1) and changes to these assumptions can have strong effects on the results, see [21, 28] for details. Since a transport modeling approach is used, not all physical aspects of the electricity transmission and $H_2$ pipeline infrastructure could be incorporated, notably the intrinsic ability of $H_2$ pipelines to buffer and store gas through linepack, although this effect is estimated small in comparison to natural gas pipelines [52]. The ESCs are modeled stand-alone and without consideration for existing infrastructure and potential options for reuse and synergies thereof. This especially includes potentially beneficial interactions with the European energy system are beyond the scope of this study and have been examined in detail by others, for example by Hess (2018) [48] and Wetzel et al. (2023) [53].

With the LCOE estimated to be below the average non-household consumer prices for electricity in France, which were at 120 EUR$_{2022}$/MWh in 2022 [54], the analysis provides grounds for considering the presented scenarios as potentially economically attractive projects. While today's electricity demand of CERN in particular, is sufficient to take off only a fraction of the electricity import volumes discussed here, future expansion and upgrades to the research facilities could lead to an increase of their electricity demand. Expansion and upgrades provide opportunities to evaluate and increase the demand-side flexibility with the potential economic benefits as outlined here. Demand-side flexibility in this study is characterized in a simplified way by assuming full demand-side flexibility capacity at zero associated costs. In reality, making available and using demand-side flexibility might be linked to costs along with the need for technical as well as organizational preparations to research equipment and usage patterns, which may dictate additional usage constraints beyond the flexibility capacity.

With the stated limitations it should become clear that this study does not constitute a feasibility study. With the focus on the techno-economic cost perspective, other aspects including societal and ecological are not explored. A more detailed investigation should incorporate an interdisciplinary approach to also evaluate to fill these gaps and investigate the social, ecological, and political consequences of detailed import projects and contain a site-specific analysis in MA or TN compared to the generalized, near country-wide analysis conducted here.

## Conclusions

The presented study analyzed importing electricity from RES from Northern Africa to Central Europe directly by HVDC transmission line and indirectly by $H_2$ pipeline. From a cost perspective, direct imports of electricity are more attractive than indirect imports by $H_2$ pipeline. The costs for electricity imports by HVDC transmission lines are estimated to be between 58 EUR/MWh to 106 EUR/MWh. Imports by $H_2$ pipeline are estimated to cost more than twice as much, between 157 EUR/MWh to 201 EUR/MWh. Demand-side flexibility for electricity can

reduce the costs of imported electricity by up to 45%, as the demand can adapt to the weather-dependent availability of the RES supply. The less flexible the electricity demand is, the more surplus infrastructure and storage capacities are required. Using CSP with TES in addition to wind and PV for the electricity mix can reduce costs further, as it allows for the use and deployment of comparatively low-cost, integrated TES as alternative energy storage to battery and $H_2$ storage. This study does not evaluate the feasibility of the proposed projects and does not assess the demand-side flexibility of research institutions. It shows that electricity from RES can be imported cost-effectively and are a possible way for Central European research institutions to reduce their energy-related scope 2 carbon emissions. In times of rising and uncertain electricity prices, research institutions could step forward to become partners and guaranteed off-takers in a project importing electricity from Northern Africa, thereby reducing project risks and securing for themselves a supply of low-cost and sustainable electricity.

## Supporting information

**S1 Appendix. Tabular results of Levelised Cost of Electricity.**
(PDF)

**S2 Appendix. Technology assumption details.**
(PDF)

**S3 Appendix. Extended and additional figures.**
(PDF)

**S4 Appendix. Regions and land availability analysis results.**
(PDF)

## Acknowledgments

The author extends his special gratitude to the following persons for their helpful and dearworthy comments and suggestions: Leon Schumm, Martina Hampp, Michael Düren and Tom Brown.

## Author Contributions

**Conceptualization:** Johannes Hampp.

**Data curation:** Johannes Hampp.

**Formal analysis:** Johannes Hampp.

**Investigation:** Johannes Hampp.

**Methodology:** Johannes Hampp.

**Project administration:** Johannes Hampp.

**Resources:** Johannes Hampp.

**Software:** Johannes Hampp.

**Validation:** Johannes Hampp.

**Visualization:** Johannes Hampp.

**Writing – original draft:** Johannes Hampp.

**Writing – review & editing:** Johannes Hampp.

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
