## [Decision Letter · Decision Letter 0]

24 Apr 2023

PONE-D-23-09476Flexing with lines or pipes: Techno-economic comparison of renewable electricity import options for European research facilitiesPLOS ONE

Dear Dr. Hampp,

Thank you for submitting your manuscript to PLOS ONE. After careful consideration, we feel that it has merit but does not fully meet PLOS ONE’s publication criteria as it currently stands. Therefore, we invite you to submit a revised version of the manuscript that addresses the points raised during the review process.

We look forward to receiving your revised manuscript.

Kind regards,

Sani Isah Abba, PhD

Academic Editor

PLOS ONE

Journal Requirements:

2. We note that Figure 2 in your submission contain [map/satellite] images which may be copyrighted. All PLOS content is published under the Creative Commons Attribution License (CC BY 4.0), which means that the manuscript, images, and Supporting Information files will be freely available online, and any third party is permitted to access, download, copy, distribute, and use these materials in any way, even commercially, with proper attribution. For these reasons, we cannot publish previously copyrighted maps or satellite images created using proprietary data, such as Google software (Google Maps, Street View, and Earth). For more information, see our copyright guidelines: http://journals.plos.org/plosone/s/licenses-and-copyright.

Reviewers' comments:

Reviewer's Responses to Questions

**Comments to the Author**

1. Is the manuscript technically sound, and do the data support the conclusions?

Reviewer #1: Yes

Reviewer #2: Yes

Reviewer #3: Partly

2. Has the statistical analysis been performed appropriately and rigorously? 

Reviewer #1: No

Reviewer #2: Yes

Reviewer #3: Yes

3. Have the authors made all data underlying the findings in their manuscript fully available?

Reviewer #1: No

Reviewer #2: Yes

Reviewer #3: Yes

4. Is the manuscript presented in an intelligible fashion and written in standard English?

Reviewer #1: Yes

Reviewer #2: Yes

Reviewer #3: Yes

5. Review Comments to the Author

Reviewer #1: The manuscript entitled “Flexing with lines or pipes: Techno-economic comparison of renewable electricity import options for European research facilities” conducts a techno-economic analysis of long-distance electricity imports from variable renewable energy sources. It also compares direct imports of electricity by High Voltage Direct Current transmission line with indirect imports by hydrogen pipelines. It is commendable that the authors included a wide range of perspectives and factors in the study. Meanwhile, extensive revision is necessary to make it suitable for publication. Here are some further remarks in more detail.

Introduction

* The background of the study is well presented. However, some literature needs to be reviewed to aid identify what has been done, what is being done currently, and what needs to be done in relation to this research area.

* Provide more information in the introduction section, on how renewable energy sources contribute to clean energy. In this sense, the following materials may be useful:

https://doi.org/10.1016/j.biteb.2022.101167

https://doi.org/10.1016/j.nexus.2022.100157

* The novelty of this paper should be presented concisely and clearly.

Method

* Transparency and replicability are two of the most crucial characteristics of good modern science, but the findings of this paper - as they currently stand, cannot be replicated because the authors have not provided enough information about their modelling techniques, input data, and underlying assumptions. For instance, what were the economic criteria used to appraise the proposed power project? Is it lifecycle cost, annualised cost, or levelised cost of energy? Furthermore, the technical criteria for appraising the proposed project should be detailed in the paper. Both technical and economic models used should be expressed using necessary mathematical equations. On this note, I recommend to you these papers for various technical and economic metrics for analysing the performances of renewable energy systems.

(i) https://doi.org/10.3390/pr8111381

(ii) https://doi.org/10.1007/s41660-022-00223-9

* Detailed capacities/ratings of components (i.e. battery, CSP, PV, wind turbines, TES, inverter, etc.) and their models with appropriate justifications for their choice should be specified.

* An overview of the software tool (PyPSA and atlite) used in this research should be presented.

Results and discussion

The authors have ample room to present and analyse the outcomes of their model, in relation to the larger body of literature. That is, the results and discussion section should be strengthened in the context of the related literature.

Fig.3 is too low-resolution and blurry. The quality of the figure should be improved.

References

* Citation is improperly done. References are not arranged in an orderly manner. For instance, the first paper cited in the introduction section has a reference number [29]. The authors are advised to follow the journal’s author guidelines for revision.

Reviewer #2: Overall the paper is of good quality. However, before approval needs some changes.

1. Abstract should be made more sound by including objective, policy implications, data period etc.

2. Introduction should be made more smooth by removing redundant lines.

3. Literature gap and contribution should also be highlighted at the end of introduction.

4. Methods must also include hypothesis which are tested in the analysis.

5. Results must also discuss in the light of past empirical and theoretical findings

6. Conclusion must also include policy suggestions.

Reviewer #3: Thank you for the opportunity to peer-review this paper. The paper uses a techno-economic analysis to compare the feasibility of direct imports of electricity by High-Voltage Direct Current transmission line with indirect imports by hydrogen pipeline for long-distance electricity imports from North Africa to Central Europe. The benefits and trade-offs of demand-side flexibility and Concentrated Solar Power with Thermal Energy Storage on the system storage requirements are also investigated.

The following observations have been raised;

1. The word “Longdistance” should be separated in the abstract.

2. The abstract is not informative, it should be improved, and numerical results clearly stated.

3. In the introduction, instead of focusing on novelty and originality, the literature review should be enhanced and updated. It is important to explain how the proposed approach adds value compared to existing research. Additionally, the study's relevance and significance should be established by identifying the limitations of previous publications that this research aims to address. Please refer to:

https://doi.org/10.3390/math11051213

https://doi.org/10.1016/j.desal.2023.116376

https://doi.org/10.1016/j.jhydrol.2020.124974

4. Study area is poorly described, rather it is unclear. Why are these stations selected? What are their particularities? More details is necessary.

5. A lot of points have been raised in the first paragraph of the Introduction but not many citations to support them. No reference was given for the “Morocco-Uk Power Project” mentioned.

6. Provide a more detailed explanation of the research methodology and data analysis techniques used in the paper.

7. Include a more comprehensive literature review to provide a better context for the research question and its significance.

8. In the Methods section, 2.1 General, the first paragraph should be rephrased to make the point clearer.

9. In text citations of the figures were written as ‘Figure’ in full instead of ‘Fig’ as shown in the guidelines.

10. In subsection 2.4, Table 1, the second column, the first letter of ‘investment’ should be capitalized.

11. The author should provide a more detailed discussion of the implications and potential impact of the research findings.

12. Collaborate with experts in related fields to provide a more interdisciplinary perspective on the research question.

6. PLOS authors have the option to publish the peer review history of their article (what does this mean?). If published, this will include your full peer review and any attached files.

Reviewer #1: **Yes: **Usman Alhaji Dodo

Reviewer #2: **Yes: **Dr Ahmed Usman

Reviewer #3: No

<quillbot-extension-portal></quillbot-extension-portal>

---

## [Author Response · Author response to Decision Letter 0]

7 Sep 2023

See dedicated document uploaded as "Response to Reviewers"

---

## [Editor Report · Decision Letter 1]

2 Oct 2023

Flexing with lines or pipes: Techno-economic comparison of renewable electricity import options for European research facilities

PONE-D-23-09476R1

Dear Dr. Hampp,

We’re pleased to inform you that your manuscript has been judged scientifically suitable for publication and will be formally accepted for publication once it meets all outstanding technical requirements.

Kind regards,

Sani Isah Abba, PhD

Academic Editor

PLOS ONE
---

## [Editor Report · Acceptance letter]

10 Oct 2023

PONE-D-23-09476R1 

Flexing with lines or pipes: Techno-economic comparison of renewable electricity import options for European research facilities 

Dear Dr. Hampp:

I'm pleased to inform you that your manuscript has been deemed suitable for publication in PLOS ONE. Congratulations! Your manuscript is now with our production department. 

Kind regards, 

on behalf of

Dr. Sani Isah Abba 

Academic Editor

PLOS ONE